# Development of Roselle (*Hibiscus sabdariffa* L.) Transcriptome-Based Simple Sequence Repeat Markers and Their Application in Roselle

**DOI:** 10.3390/plants13243517

**Published:** 2024-12-16

**Authors:** Aifen Tao, Yunqing Li, Jihan Chen, Jing Li, Jiantang Xu, Lihui Lin, Liwu Zhang, Pingping Fang

**Affiliations:** 1Key Laboratory of Ministry of Education for Genetics, Breeding and Multiple Utilization of Crops, Fujian Agriculture and Forestry University, Fuzhou 350002, China; aifentao@fafu.edu.cn (A.T.); cnfiber@126.com (J.X.); lihui9027@163.com (L.L.); lwzhang@fafu.edu.cn (L.Z.); 2Fujian Key Laboratory of Crop Breeding for Design, Fujian Agriculture and Forestry University, Fuzhou 350002, China; 3Yantai Institute, China Agricultural University, Yantai 264000, China; lyq17853514030@163.com; 4School of Agriculture, Northeast Agricultural University, Harbin 150030, China; jihanchen09@163.com; 5State Key Laboratory of Tree Genetics and Breeding, Beijing Forestry University, Beijing 100083, China; jingli1396@163.com

**Keywords:** *Hibiscus sabdariffa* L., simple sequence repeat markers, transcriptome-based, application

## Abstract

Roselle (*Hibiscus sabdariffa* L.) simple sequence repeat (SSR) markers were developed using RNA sequencing technology, providing a foundation for genetic analysis and the identification of roselle varieties. In this study, 10 785 unigenes containing 12 994 SSR loci with an average of one SSR locus per 6.87 Kb were identified, and the occurrence frequency of the SSR loci was 11.36%. Trinucleotide repeat motifs were the most abundant, followed by dinucleotide repeats, with AAG/CTT and AT/AT being the predominant types, respectively. After screening 100 primer pairs with a polymorphic ratio of 32.0%, we obtained 32 primer pairs, resulting in clear and stable polymorphic bands. Twenty-seven primer pairs were highly or moderately polymorphic, and seven primer pairs were highly polymorphic. Genetic relationship analysis based on the selected SSR primers showed that 38 roselle accessions were classified into different clades, with those from the same regions clustered into the same subgroups. In contrast, individuals with unique morphological traits were separated. DNA fingerprints of 38 roselle varieties were constructed using five SSR primers, providing an effective method for identifying roselle varieties at a molecular level. Our data provide novel insights into the genetics of *H. sabdariffa* and may be used in SSR-assisted roselle breeding.

## 1. Introduction

Roselle (*Hibiscus sabdariffa* L.) comprises two subspecies, *Hibiscus sabdariffa* var. *sabdariffa* (HSS) and *Hibiscus sabdariffa* var. *altissima* (HSA), and is an essential annual crop planted in the tropics and subtropics [1]. HSS is edible, whereas HSA is commercially used as an alternative fiber source [2]. Roselle calyx has high nutritional, medicinal, and economic value [3,4]. Besides being consumed as jam, syrup, and drinks, it is also used as a natural food colorant [5]. Furthermore, roselle is widely used in the food, beverage, and pharmaceutical industries because of its high anthocyanin and vitamin C content [6].

Investigating genetic diversity helps to reveal the evolutionary history of species or populations, predict adaptability, and estimate the distribution of genetic resources, which is crucial for collecting, preserving, evaluating, and using germplasm resources [7]. Roselle varieties cultivated in China were introduced from abroad in the 1910s [8]. Owing to this and the renaming in different regions, the genetics of roselle germplasm in China remained uncertain, which heavily restricted preservation, parental selection, and breeding development. Thus, elucidating genetic variation and sustainability can aid in the genetic improvement and molecular breeding of roselle.

DNA fingerprinting is a powerful tool for identifying crop varieties and strains at the molecular level and can provide technical support for variety identification and protection [9]. Furthermore, using molecular marker technology to construct DNA fingerprints can help to avoid repetitive collection and preservation of crop germplasm [10].

Usually, evaluating phenotypic traits is a common approach for assessing crop diversity. However, these investigations are expensive and inefficient for the assessment of complex polygenic traits [11]. In contrast, DNA-based markers provide reliable results regardless of the external environment and can resolve these issues [12]. Among the DNA-based molecular markers, simple sequence repeat (SSR) markers are highly polymorphic and exhibit co-dominant inheritance, extensive distribution within the plant genome, and are transferrable among closely related species. They offer valuable insights into genetic relationships, population structure, and genetic diversity, and are usually the preferred choice in studies across diverse plant species [13,14,15]. SSRs are essential molecular techniques used in breeding programs and for discovering important traits for crop improvement [16,17]. Furthermore, SSR markers together with SNP markers are recommended by the International Union for the Protection of New Plant Varieties as molecular markers for constructing DNA fingerprint databases [18] and have been widely used in crops, such as maize [19], rice [20], and wheat [21].

Because SSR markers are a reliable and powerful tool for identifying crop germplasm and analyzing genetic diversity [22,23], obtaining suitable SSR markers for roselle is essential. However, traditional SSR development methods are time-consuming and inadequate for meeting growing research demands. Transcriptome sequencing can quickly, comprehensively, and accurately reveal genetic information and is efficient for developing molecular markers that are widely used in genetic research and molecular breeding [24,25]. With the development of transcriptomics and sequencing technologies, SSR markers based on next-generation sequencing data have been widely applied to crops, including maize [26], sweet potato [27], soybean [28], and orchids [29]. Expressed sequence tag (EST)-based SSR markers can be detected more accurately, rapidly, and efficiently than traditional and genomics-based SSR markers [30]. Obtaining SSR markers using transcriptomic data is an effective approach for non-model crops such as roselle. However, few EST-based SSR markers have been developed for *H. sabdariffa*, which limits research on germplasm identification, genetics, and breeding of roselle at the molecular level. In this study, a de novo transcriptome assembly of the RNA-sequencing library of *H. sabdariffa* was obtained, and a number of SSR markers were developed from the transcript assemblies. Furthermore, polymorphisms were identified in randomly selected SSR markers, which were used for genetic diversity analyses and DNA fingerprinting of roselle accessions. This study provides a valuable and efficient tool for genetic diversity analysis, variety identification, and marker-assisted genetic breeding in roselle.

## 2. Results

### 2.1. Transcriptome Assembly of Roselle

In total, 6.25 Gb of raw data were obtained for roselle transcriptome assembly based on 100 bp paired-end reads, available in the NCBI database (https://submit.ncbi.nlm.nih.gov/) under PRJNA885398. In this study, 66,925 unigenes with an average length of 780 bp and a GC content of 45.51% were obtained (Table 1). The length and quantity of the N50 unigenes were 19,260 and 1325, respectively, indicating relatively stable quality. A total of 1077 genes of complete Benchmarking Universal Single-Copy Orthologs (BUSCOs) and 977 genes of complete and single-copy BUSCOs were screened using BUSCO evaluation. Furthermore, 80 complete and duplicated BUSCOs, 207 fragmented BUSCOs, and 156 missing BUSCOs were identified.

Additionally, 65,762 unigenes obtained from the assembly were annotated using the Non-Redundant Protein Sequence Database. A total of 47,076 unigenes were divided into 25 functional categories, including general function prediction, signal transduction mechanisms, post-translational modification, and protein turnover using Clusters of Orthologous Groups (COGs) database annotations. Gene ontology analysis categorized these roselle unigenes into molecular functions, cellular components, and biological processes. Furthermore, 63,073 assembled unigenes were annotated in the Kyoto Encyclopedia of Genes and Genomes (KEGG) database, accounting for approximately 55% of the total genes. These results indicate that roselle sequence data may provide a useful reference for molecular research in *Hibiscus* plants.

### 2.2. Characteristics of the SSR Loci in H. sabdariffa

In total, 12,994 SSR loci distributed across 10,785 unigenes were identified in 66,925 *H. sabdariffa* unigenes. Of these, 1121 had more than two SSR loci, whereas 9664 contained a single SSR locus. The occurrence frequency of SSR loci was 11.36%, with an average of one SSR loci per 6.87 Kb. *H. sabdariffa* SSRs from the transcriptome contained different repeat types, including di-, tri-, tetra-, penta-, and hexanucleotides, but not mononucleotides. The frequency of trinucleotides was the highest (40.17%), followed by dinucleotides (35.20%) and tetranucleotides (11.38%). The percentages of pentanucleotide (6.34%) and hexanucleotide (6.91%) repeats were low (Table 2). These data indicate that shorter repeats (dinucleotides and trinucleotides) had a higher mutational frequency in *H. sabdariffa*.

The dominant repeat type in the dinucleotide motif was AT/AT (1995 repeats; 43.60%) (Figure 1), followed by AG/CT repeats (40.55%). Eight types of repeats were observed in the trinucleotide motif, with AAG/CTT being predominant (1841 repeats; 35.27%). AAAG/CTTT was the most abundant tetranucleotide motif (409 repeats; 27.65%). The other motif levels, such as pentanucleotides and hexanucleotides, were low. A and T were rich in the most predominant type of each repeat motif, and C base methylation might account for this result [31].

### 2.3. SSR Primer Validation

Initially, eight individual roselle accessions were randomly screened using 100 primer pairs to validate amplification efficacy, and 67 pairs successfully produced clear bands. These 67 primer pairs were selected again using DNA from 38 individual plant collections as templates. Thirty-two primer pairs resulted in clear, stable, and polymorphic bands (Figure 2) with a total polymorphic ratio of 32.0%. Among the 32 selected primer pairs, the dinucleotide repeat type was the most abundant (19), followed by trinucleotides (9). Tetranucleotides were the least abundant, with a count of four. The size of the amplified sequences ranged from 153 to 278 bp. Details of the selected SSR primers are shown in Appendix A.

A total of 293 bands were amplified in 38 accessions using 32 primer pairs, yielding 286 polymorphic bands with a ratio of 97.6%. A total of 209 alleles, with an average of 6.53 alleles (3–12 alleles) per locus, were detected (Table 3). Among the 32 primer pairs, the effective number of alleles per locus varied from 1.246 to 4.356, with primer SSR45 containing the most effective alleles (4.356 alleles), followed by SSR76 (3.592 alleles) and SSR77 (3.156 alleles). Expected and observed heterozygosity values were 0.2–0.781 (SSR45) and 0.026–1.000 (SSR17, SSR62), respectively. Shannon’s information index (I) was 0.474–1.839 (SSR45), Nei’s information index (H) was 0.197–0.770 (SSR45), and polymorphism information content (PIC) values varied from 0.194 to 0.750, with a mean of 0.408. PIC indicates the level of SSR polymorphism. SSR marker polymorphism can be divided into three intervals according to PIC values: low (<0.25), medium (0.25 > PIC > 0.50), and high (>0.50). Among these primer pairs, seven were classified as highly polymorphic (PIC > 0.5) (SSR36, SSR45, SSR58, SSR59, SSR76, SSR77, and SSR83), 20 as moderately polymorphic (0.25 < PIC < 0.50), and five were low polymorphic (PIC < 0.25). All seven highly polymorphic primers belonged to the dinucleotide (five) and trinucleotide types (two).

### 2.4. Genetic Diversity Analysis of 38 Roselle Accessions

The genetic similarity coefficient among the 38 germplasms varied from 0.30 to 0.84, with an average value of 0.42. This result indicates that the genetic relationships of the 38 roselle accessions are relatively distant, with considerable differences. The 38 accessions were classified into three main clades and one separate branch, according to the neighbor-joining method (Figure 3). Group II was the largest, containing 30 individuals, followed by Group I, with five individuals. Group III had only two accessions (No. 37 and 38) from Zhejiang Province. Germplasm No. 29, collected from Hunan Province, was grouped into a separate branch (Group IV), consistent with its unique and specific morphological characteristics (Figure 4). No. 27, which had alternating red and white calyces, was separated from the individuals with red calyces in Group II (Figure 5). Germplasms No. 6, 7, 8, and 9 from Hunan Province were clustered into a subgroup belonging to HSA. No. 1 and 2 (from Fuzhou, Fujian); 17, 18, and 19 (from Zhangzhou, Fujian); 25 and 26 (from Guangdong); 37 and 38 (from Zhejiang); and 15, 16, 20, and 28 (from Fujian) were divided into different subgroups according to their geographical distributions. Some germplasm lines collected from different areas clustered together, such as No. 10 (from Hunan), 11 (from Myanmar), 22 (from Yunan), and 23 (from Jiangxi), as well as 34 (from Fujian) and 35 (from Zhejiang). This inconsistency may have been the result of germplasm introduction from different regions.

### 2.5. DNA Fingerprinting of Roselle Accessions

DNA fingerprinting aimed to distinguish a considerable number of varieties with minimal primers, and only five SSR primer pairs (SSR45, SSR58, SSR59, SSR76, and SSR77) were required to differentiate all 38 roselle accessions. The results of ‘0’ and ‘1’ generated using the five SSR primer pairs were combined into a series of numbers through DNA amplification of the 38 roselle varieties (Appendix A), and the specific digital fingerprints of each roselle variety were established. A QR (Quick Response) code for each variety was generated, and the variety names, collection regions, and DNA fingerprinting information were obtained (Figure 6). The constructed DNA fingerprints can identify differences in DNA levels between different roselle cultivars, providing a theoretical basis for identifying roselle varieties at the molecular level.

## 3. Discussion

### 3.1. Characteristics of the Developed SSR Loci in Roselle

Developing SSR markers from transcriptome sequences is essential for identifying the association between functional genes and phenotypes, especially in non-model species [30]. The developed transcriptome-based SSR markers have higher specificity and better inter-species universality because most transcriptome sequences are concentrated in functional genes, and the sequences are sufficiently conserved. The development and use of transcriptome-based SSR markers have been widely performed in the *Malvaceae* family, including kenaf (*Hibiscus cannabinus* L.) [32], cotton (*Gossypium hirsutum* L.) [33], hibiscus (*Hibiscus syriacus* L.) [34], and okra (*Abelmoschus esculentus*) [35]. Additionally, SSR markers developed through plant transcriptome sequences have been used for the assessment of population genetic diversity and phylogenetic relationships in endangered species, such as *Meconopsis integrifolia* (Maxim.) Franch. and *Saussurea involucrate* [13,15], which provided valuable information and a scientific basis for the conservation, systematic utilization, and sustainable management of some endangered but economically and ornamentally significant plants.

In our study, the frequency of the roselle transcriptome-based SSR loci was 11.36%, which was lower than that of okra (20.59%) [35], kenaf (18.07%) [36], and papaya (*Chaenomeles sinensis*) (7.48%) [37]. The distribution frequency of SSR loci varies greatly among species, which may be caused by the genomic size, tissue, stage of the plants used for transcriptome sequencing, screening criteria, sequencing depth, and assembly quality [38].

A previous study reported that trinucleotide repeats are the most common motifs in mono- and dicotyledonous plants [27]. In this study, trinucleotide repeat motifs were the most abundant SSRs, consistent with the results of kenaf [32], okra [35], and cotton [39]. This phenomenon is probably because trinucleotides are more stable than other repeat unit types in the protein-coding region, rarely producing frameshift mutations in the coding frame, which is consistent with longer repeat sequences being less stable due to higher mutation rates [40]. However, SSR markers developed based on the jute (*Corchorus* L.) transcriptome are primarily mononucleotide repeats [41] and are similar to those of sisal (*Agave sisalana* Perr. ex Engelm.) [42], *Apocynum* (toina) [23], and rose (*Rosa rugosa* Thunb.) [43]. The variance in the abundance and distribution of SSR motifs can be explained by differences in species, SSR detection criteria, dataset size, sequence redundancy, and types of data-mining tools [44].

The dinucleotide repeat type AT/AT was the most dominant motif, followed by the AG/CT type. AAG/CTT had the highest distribution frequency, consistent with the results of studies on cotton [39], okra [35], jute [41], and sweet potatoes (*Ipomoea batatas* [L.] Lam.) [27]. Additionally, trinucleotides contain various motif types with considerable differences in their distribution frequencies. Studies on *Arabidopsis* and soybean (*Glycine max* [L.] Merr.) have indicated that the trinucleotide AAG motif is common in dicotyledonous plants [45,46].

### 3.2. The Polymorphisms of SSR Primers Developed Through Roselle Transcriptomes

In the present study, the 32 SSR primer pairs with high polymorphisms in 38 roselle accessions were identified using 100 primer pairs, with a polymorphism rate of 32%. The SSR polymorphism rate of roselle is higher than that of jute (27.58%) [41], but lower than that of *Hibiscus aridicola* (47.6%) [34], okra (60%) [35], kenaf (61%) [36], and cotton (65%) [39]. The polymorphism level of SSR primers in roselle was moderate and effective [47,48,49,50]. The SSR markers developed based on the roselle transcriptome data had an ideal polymorphism rate. Additionally, the polymorphism rate varies among species, which may be related to the degree of DNA transcription sequence conservation among them [41].

PIC values of the 32 SSR marker pairs used in this study ranged from 0.194 to 0.750, with an average of 0.408. This value is higher than the average PIC value in 124 roselle samples (0.32) [2], which may be because the SSR markers used in previous studies were not developed from the roselle transcriptome or genomic data. Meanwhile, the PIC value in this study was substantially higher than the average value for okra (0.12) [35]. Additionally, observed heterozygosity (Ho) and expected heterozygosity (He) values indicate the magnitude of genetic variance for different SSR primers, with higher Ho values indicating higher heterozygosity [51]. In the present study, 16 out of the 32 investigated loci showed higher heterozygote deficiency compared to the HWE (Hardy–Weinberg equilibrium) expectations. This situation is commonly observed in studies using SSR markers and is due to the occurrence of “silent” alleles caused by the lack of amplification of certain fragments [52]. In summary, the newly developed SSR primers in roselle showed high polymorphism, and the roselle individuals used in this study exhibited wide genetic diversity.

### 3.3. Genetic Diversity of 38 Roselle Varieties

The phylogenetic relationship of different germplasms is essential and may provide insights into their genetic correlations, facilitating the further use of genetic resources. To the best of our knowledge, most roselle cultivars planted in China have been introduced from abroad, and due to regional introduction and hybrid breeding, their genetic backgrounds are relatively complex and unclear. Furthermore, few reports have assessed the genetic diversity in roselle germplasms using SSR markers [2], which hinders genetic and breeding research on roselle. However, the SSR markers developed in this study were able to cluster the 38 roselle germplasms into different groups and subgroups according to their agronomic characteristics or geographical relationships. For instance, some accessions (No. 27, 29, and 6–9) with special traits in leaf shape, flowering time, and calyx color were identified using SSR clustering analysis, which could facilitate the exploration of a large number of candidate genes associated with important traits. Meanwhile, individual plants may exhibit slight but discernible differences in their growth period, anthocyanin content, and stress tolerance. The genetic traits underlying these characteristics may be associated with SSR markers, providing a scientific foundation for describing these characteristics [53]. In addition, few individuals from different areas clustered into the same subgroup, which may be due to germplasm introduction among different regions. Overall, this diversity assessment will aid in selecting the trait-specific genotypes for crop improvement programs, marker-assisted breeding, and quantitative trait loci identification for roselle.

### 3.4. DNA Fingerprinting of Roselle Accessions

Identifying crop varieties is crucial because of the increasing exchange of germplasm resources locally and internationally [54]. DNA fingerprinting is an effective tool for confirming and distinguishing between different varieties, which has the characteristics of uniqueness and identifiability. In particular, QR codes effectively accommodate numbers, characters, pictures, and other information and can be quickly recognized by computers, mobile phones, and other electronic devices [9]. These codes are convenient and intuitive in presenting cultivar-related information and have been widely used to rapidly identify crop varieties [55]. We constructed unique and cultivar-specific DNA fingerprints of the 38 roselle germplasm lines.

Furthermore, the DNA fingerprints constructed in this study can be used to improve the management of the Roselle Germplasm Bank, providing an essential technical foundation for constructing a standardized DNA fingerprinting library of roselle germplasm. Constructing roselle molecular fingerprints plays a crucial role in identifying and protecting germplasm, facilitating gene mapping, and enabling functional research, thus maximizing the research potential of germplasm resources.

## 4. Materials and Methods

### 4.1. Plant Materials

Thirty-eight individuals from the *H. sabdariffa* species were collected, which were provided by the Key Laboratory of Ministry of Education for Genetics, Breeding, and Multiple Utilization of Crops (Fuzhou, China), Institute of Bast Fiber Crops, Chinese Academy of Agriculture Sciences (Changsha, China), and the Institute of Subtropical Agriculture, Fujian Academy of Agricultural Sciences. Details on the collection locations and cultivar names of all samples are listed in Table 4. These germplasms were cultivated in a greenhouse at the Resource Nursery of Fujian Agriculture and Forestry University (Fuzhou, China). The seeds were sown in pots containing nutrient soil, with both the diameter and height of the pots being 20 cm. The seedlings were grown under greenhouse conditions, with a temperature range of 25–35 °C (night/day), a photoperiod of 14 h, and a relative humidity of 60–80%. Fresh calyces of ‘FZ-72’ were collected 30 days post-flowering for RNA extraction and transcriptome sequencing.

### 4.2. DNA and RNA Extraction, cDNA Library Construction, and Sequencing

Genomic DNA was extracted from all specimens using a modified DNA extraction method to detect polymorphisms at isolated microsatellite loci [56]. Total RNA was obtained from the roselle calyces using the Quick RNA Isolation Kit (ZH120; Huayueyang, Beijing, China). The quality and quantity of the extracted DNA and RNA were assessed using 1.2% agarose gel electrophoresis and a NanoDrop 2000 spectrophotometer (ThermoFisher Scientific, Wilmington, DE, USA). Approximately 200 bp of cDNA was screened using AMPure XP Beads (BECKMAN COULTER Life Sciences, Shanghai, China) for PCR amplification, and the PCR products were purified using AMPure XP Beads to construct a complete sequencing library. The constructed library was sequenced on an Illumina HiSeqTM2500 platform. The assembly integrity was evaluated using BUSCO (http://busco.ezlab.org/) (Version 3.0.2, accessed on 12 November 2021).

### 4.3. Transcriptome Assembly and Unigene Annotation

Raw data were filtered by trimming the adaptor sequences, and low-quality sequences (Q < 20) containing >10% of uncertain (N) bases were removed. Trinity was used to conduct the de novo assembly [57]. The transcripts were assembled, and the main transcript was selected from the local area as a unigene [58]. The assembled unigene sequences were compared with the ones from protein databases NR, SwissProt, KEGG, and COG/KOG, and the protein with the highest sequence similarity to the given unigene was used to obtain functional annotation information [59].

### 4.4. Transcriptome Data-Based Microsatellite Identification

MicroSatellite software v.2.1 (MISA; http://pgrc.ipk-gatersleben.de/misa/misa.html, accessed on 25 February 2022) was used to identify SSR loci [60]. Sequences with mono-, di-, tri-, tetra-, penta-, and hexanucleotide units were repeated at least fifteen, eight, five, four, three, and two times, respectively. The flanking sequence length at the SSR sites was >150 bp. Furthermore, 100 SSR primers were designed using Primer Premier 5.0 (www.bio-soft.net) with the following parameters: a primer length of 20–40 bp, a PCR product size of 150–300 bp, an annealing temperature of 55–60 °C, and a GC content of 40–60%. All primers were synthesized by Qingke Biotech (Guangzhou, China).

### 4.5. SSR Marker Evaluation and Genetic Diversity Analysis of Roselle

Eight germplasm samples from 38 collections were amplified to detect SSR efficiency and polymorphisms. All 38 individuals were amplified using polymorphic SSR primers to identify relationships. PCR was performed using a Bio-Rad T100™ Thermal Cycler (Bio-Rad, Hercules, CA, USA) in a total volume of 20 µL containing 10 µL 2 × Taq Master Mix, 0.5 μM of each primer, and 50 ng of genomic DNA. The reaction conditions were as follows: 3 min of denaturation at 94 °C, 35 cycles of denaturation at 94 °C for 30 s, annealing at 55–60 °C for 30 s, elongation at 72 °C for 1 min, and a 5 min final elongation at 72 °C. The PCR products were separated using 6% non-denaturing polyacrylamide gel electrophoresis. Clear bands from the PCR products were genotyped using 0.1% silver staining [61]. DNA bands generated using the SSR markers were compared with the ones of standard molecular weight markers, and ‘1’ and ‘0’ were used to indicate the presence and absence of a band, respectively. Popgene v.1.32 (https://sites.ualberta.ca/~fyeh/popgene.html, accessed on 6 March 2022) was used to calculate multiple indices of marker efficiency, including the number of alleles, expected heterozygosity, observed heterozygosity, Shannon’s information index (I), and Nei’s information index (H). PIC was generated using PowerMarker v.3.25 (https://brcwebportal.cos.ncsu.edu/powermarker/, accessed on 10 March 2022). Finally, the genetic similarity coefficients of the 38 roselle germplasm lines were calculated using NTSYSpc v.2.1 (https://www.appliedbiostat.com/ntsyspc/ntsyspc.html, accessed on 12 March 2022), and a genetic clustering diagram was constructed [62].

### 4.6. DNA Fingerprinting and QR Code Construction for Roselle Varieties

Five SSR primer pairs with ideal polymorphisms were selected for PCR amplification across the 38 roselle varieties, and DNA fingerprints were constructed as previously described [63]. The amplification bands at different allele loci were marked as ‘1’, whereas bands without amplification were marked as ‘0’. Subsequently, QR code generator (http://cli.im/, accessed on 7 April 2024) was used to create a QR code for each variety, and the variety name, collection region, and digital fingerprints were recorded together.

## 5. Conclusions

This study developed SSR markers based on the roselle transcriptome, with an occurrence frequency of 11.36%. Trinucleotide repeat motifs were the most common repeat types. Additionally, 32 primer pairs that produced clear and stable polymorphic bands were obtained by screening 100 primer pairs; 27 of these primer pairs were highly or moderately polymorphic. The selected SSR primers were used to analyze the genetic relationships of the 38 roselle accessions, and the varieties were classified into different groups according to certain characteristics, including collection regions and morphological traits. Specific DNA fingerprints of the 38 roselle varieties were constructed using five SSR primers. This research developed an effective molecular marker technique for the identification, genetic relationship analysis, and molecular-assisted breeding of *H. sabdariffa*.

## Figures and Tables

**Figure 1 plants-13-03517-f001:**
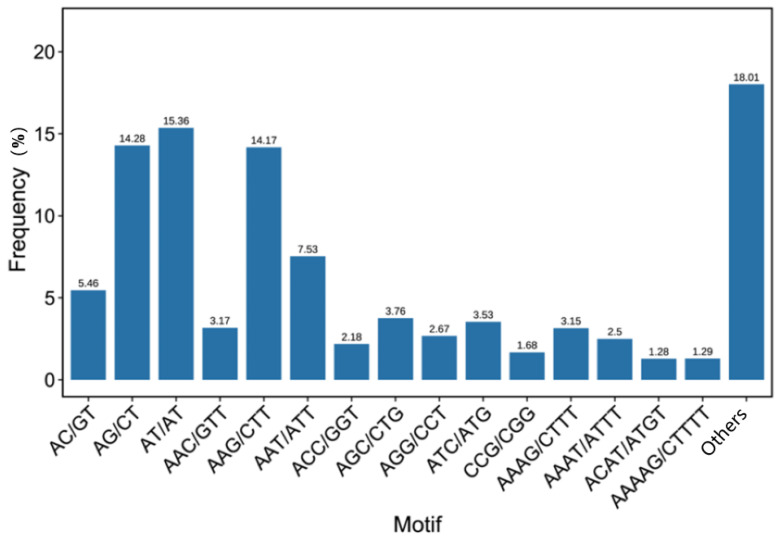
Proportion of primary SSR repeat types in the roselle SSR loci.

**Figure 2 plants-13-03517-f002:**
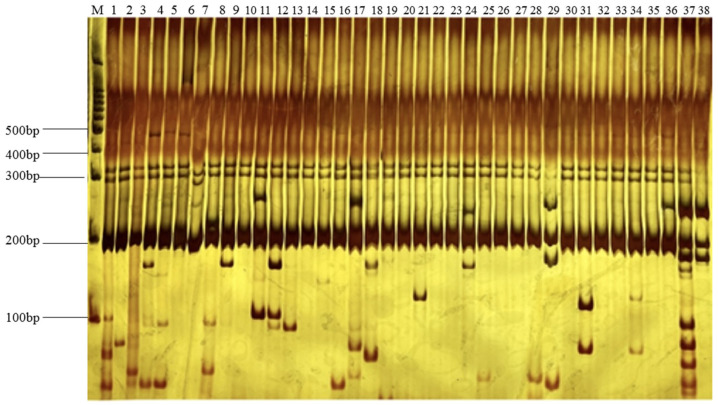
Amplification results of 38 roselle varieties using SSR45 primer pairs.

**Figure 3 plants-13-03517-f003:**
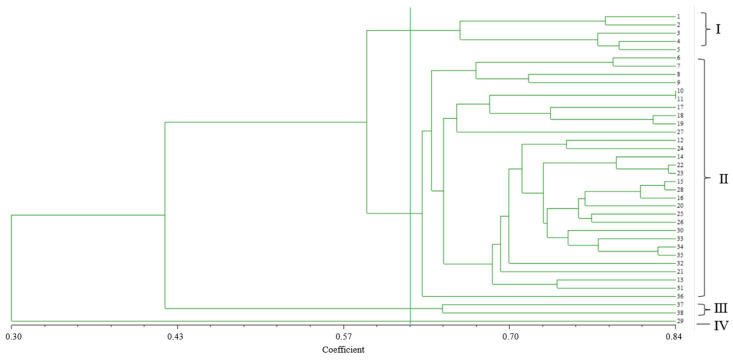
Dendrogram of the 38 roselle accessions based on SSR markers. The green vertical line indicates that the 38 accessions can be classified into four groups when the genetic similarity coefficient is 0.62.

**Figure 4 plants-13-03517-f004:**
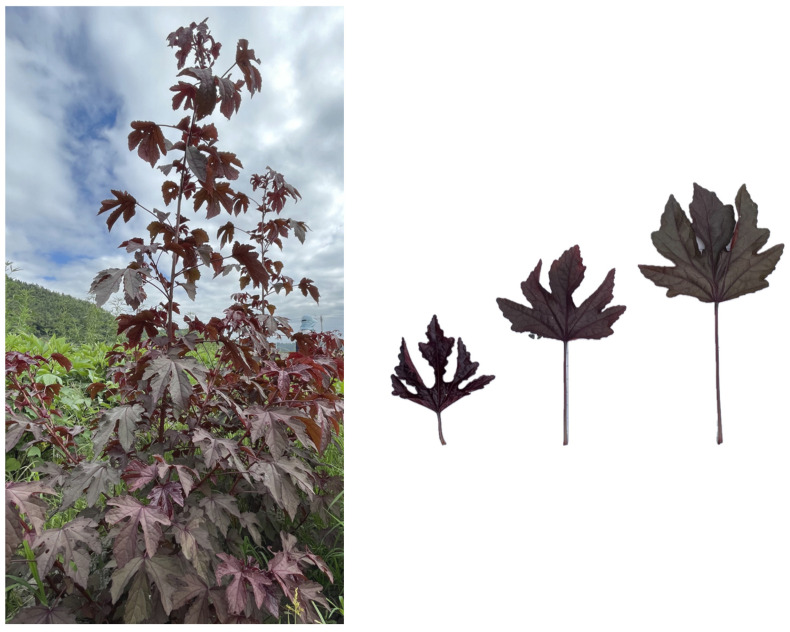
Plant No. 29 and its leaf morphology.

**Figure 5 plants-13-03517-f005:**
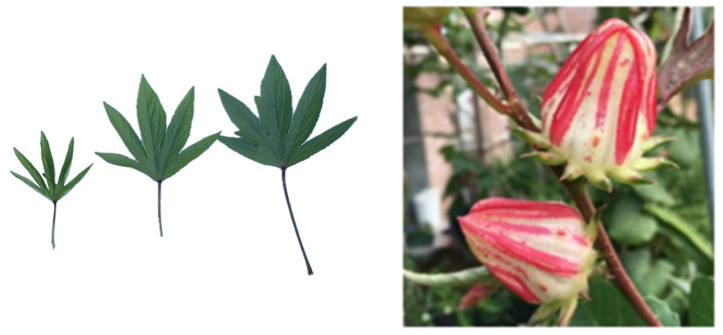
Leaf morphology and fruit of plant No. 27.

**Figure 6 plants-13-03517-f006:**
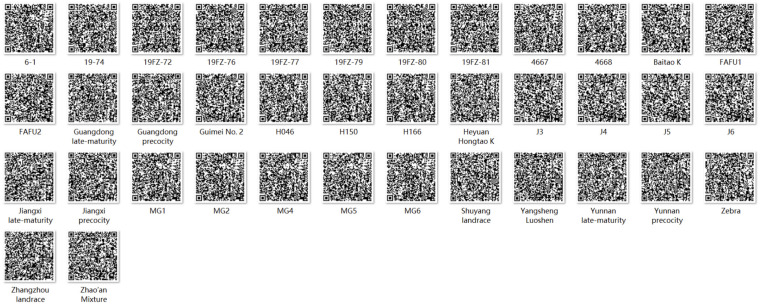
DNA fingerprints of the 38 roselle accessions.

**Table 1 plants-13-03517-t001:** Summary information of the roselle transcriptome.

Term	Number/Length
Raw reads (bp)	41,660,990
Raw bases (Gb)	6.249
Clean reads (bp)	41,530,040
Clean bases (Gb)	6.016
Valid base	96.26%
Q 30	92.45%
GC content	45.51%
Number of unigenes	66,925
N50 (bp)	19,260

**Table 2 plants-13-03517-t002:** SSR repeat types in the roselle transcriptome.

SSR Type	Number	Frequency (%)	Average Distance (Kb)
Dinucleotide repeats	4574	35.20	19.52
Trinucleotide repeats	5220	40.17	17.10
Tetranucleotide repeats	1479	11.38	60.38
Pentanucleotide repeats	824	6.34	108.37
Hexanucleotide repeats	897	6.91	99.55

**Table 3 plants-13-03517-t003:** Genetic characteristics of the 32 SSR primers in 38 roselle accessions.

Primer Code	Number of Alleles	Effective Number of Alleles/Loci	Expected Heterozygosity	Observed Heterozygosity	Shannon’s Diversity Index	Nei’s Diversity Index	Polymorphism Information Content
SSR8	7	1.483	0.330	0.316	0.768	0.326	0.314
SSR10	5	1.395	0.287	0.158	0.631	0.283	0.272
SSR15	6	1.969	0.499	0.290	1.073	0.492	0.471
SSR17	3	2.053	0.520	1.000	0.754	0.513	0.394
SSR36	6	2.429	0.596	0.026	1.221	0.588	0.554
SSR39	6	2.001	0.507	0.105	1.032	0.500	0.470
SSR42	6	1.278	0.221	0.132	0.518	0.218	0.210
SSR44	4	1.306	0.238	0.263	0.474	0.234	0.217
SSR45	10	4.356	0.781	0.132	1.839	0.770	0.750
SSR49	7	1.319	0.245	0.105	0.601	0.242	0.235
SSR50	9	1.523	0.348	0.368	0.812	0.344	0.328
SSR55	8	1.638	0.395	0.316	0.955	0.390	0.380
SSR56	8	1.489	0.333	0.211	0.811	0.328	0.320
SSR58	7	2.845	0.657	0.105	1.326	0.649	0.605
SSR59	12	2.743	0.644	0.316	1.542	0.635	0.615
SSR60	4	2.163	0.545	0.974	0.857	0.538	0.431
SSR62	4	2.222	0.557	1.000	0.899	0.550	0.448
SSR66	5	2.064	0.523	0.711	0.881	0.516	0.429
SSR74	5	1.905	0.481	0.658	0.808	0.475	0.395
SSR76	9	3.592	0.731	0.237	1.569	0.722	0.679
SSR77	7	3.156	0.692	0.237	1.425	0.683	0.642
SSR78	3	2.050	0.519	0.974	0.753	0.512	0.394
SSR80	7	1.443	0.311	0.263	0.750	0.307	0.299
SSR82	7	1.320	0.246	0.263	0.613	0.242	0.237
SSR83	4	2.845	0.657	0.184	1.192	0.649	0.594
SSR84	6	2.120	0.535	0.105	0.912	0.528	0.437
SSR87	9	1.631	0.392	0.132	0.933	0.387	0.374
SSR89	7	1.246	0.200	0.211	0.519	0.197	0.194
SSR94	10	1.796	0.449	0.421	1.069	0.443	0.428
SSR98	6	1.358	0.267	0.237	0.637	0.264	0.257
SSR99	5	1.507	0.341	0.395	0.685	0.337	0.312
SSR100	7	1.626	0.390	0.395	0.888	0.385	0.370
Average	6.531	1.996	0.451	0.351	0.930	0.445	0.408

**Table 4 plants-13-03517-t004:** Information on the 38 roselle accessions.

No.	Accession Number	Species	Collection Region
1	FAFU1	*H. sabadariffa* var. *sabdariffa*	Fujian, China
2	FAFU2	*H. sabadariffa* var. *sabdariffa*	Fujian, China
3	Heyuan Hongtao K	*H. sabadariffa* var. *sabdariffa*	Guangdong, China
4	19FZ-81	*H. sabadariffa* var. *sabdariffa*	Fujian, China
5	H046	*H. sabadariffa* var. *sabdariffa*	Hunan, China
6	H150	*H. sabadariffa* var. *altissima*	Hunan, China
7	H166	*H. sabadariffa* var. *altissima*	Hunan, China
8	4667	*H. sabadariffa* var. *altissima*	Hunan, China
9	4668	*H. sabadariffa* var. *altissima*	Hunan, China
10	Shuyang landrace	*H. sabadariffa* var. *sabdariffa*	Hunan, China
11	19FZ-74	*H. sabadariffa* var. *sabdariffa*	Myanmar
12	Baitao K	*H. sabadariffa* var. *sabdariffa*	Guangdong, China
13	Zhao’an Mixture	*H. sabadariffa* var. *sabdariffa*	Fujian, China
14	Guimei No. 2	*H. sabadariffa* var. *sabdariffa*	Guangxi, China
15	Zhangzhou Landrace	*H. sabadariffa* var. *sabdariffa*	Fujian, China
16	MG1	*H. sabadariffa* var. *sabdariffa*	Fujian, China
17	MG2	*H. sabadariffa* var. *sabdariffa*	Fujian, China
18	MG4	*H. sabadariffa* var. *sabdariffa*	Fujian, China
19	MG5	*H. sabadariffa* var. *sabdariffa*	Fujian, China
20	MG6	*H. sabadariffa* var. *sabdariffa*	Fujian, China
21	Yunnan precocity	*H. sabadariffa* var. *sabdariffa*	Yunnan, China
22	Yunnan late-maturity	*H. sabadariffa* var. *sabdariffa*	Yunnan, China
23	Jiangxi precocity	*H. sabadariffa* var. *sabdariffa*	Jiangxi, China
24	Jiangxi late-maturity	*H. sabadariffa* var. *sabdariffa*	Jiangxi, China
25	Guangdong precocity	*H. sabadariffa* var. *sabdariffa*	Guangdong, China
26	Guangdong late-maturity	*H. sabadariffa* var. *sabdariffa*	Guangdong, China
27	Zebra	*H. sabadariffa* var. *sabdariffa*	Guangdong, China
28	19FZ-80	*H. sabadariffa* var. *sabdariffa*	Fujian, China
29	19FZ-79	*H. sabadariffa* var. *altissima*	Fujian, China
30	6-1	*H. sabadariffa* var. *sabdariffa*	Fujian, China
31	19FZ-77	*H. sabadariffa* var. *sabdariffa*	Fujian, China
32	19FZ-76	*H. sabadariffa* var. *sabdariffa*	Fujian, China
33	Yangsheng Luoshen	*H. sabadariffa* var. *sabdariffa*	Hebei, China
34	19FZ-72	*H. sabadariffa* var. *sabdariffa*	Fujian, China
35	J3	*H. sabadariffa* var. *sabdariffa*	Zhejiang, China
36	J4	*H. sabadariffa* var. *sabdariffa*	Zhejiang, China
37	J5	*H. sabadariffa* var. *sabdariffa*	Zhejiang, China
38	J6	*H. sabadariffa* var. *sabdariffa*	Zhejiang, China

## Data Availability

All data generated during this study are included in this published article, and the raw data used or analyzed during the current study are available from the NCBI (https://submit.ncbi.nlm.nih.gov/) with PRJNA885398.

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
