# Peer review of "Development of Roselle (Hibiscus sabdariffa L.) Transcriptome-Based Simple Sequence Repeat Markers and Their Application in Roselle"

_plants, 2024, doi:10.3390/plants13243517_

Round 1
Reviewer 1 Report
Comments and Suggestions for Authors
Comments to the paper: “The development of roselle (Hibiscus sabdariffa L.) transcriptome-based simple sequence repeat markers and their application in roselle” by Aifen Tao et al.
The authors developed simple sequence repeat (SSR) markers based on the RNA-sequencing technology of roselle calyces. Subsequently, they used 32 of them to screen genetic variation in that 38 roselle accessions. Their results showed that that the investigated accessions could be classified into different clades, and those from the same regions clustered into the same subgroups. They also found that only five SSR primers were required to distinguish all 38 roselle accessions. Furthermore, the authors carried out gene ontology analysis (not mentioned in the Abstract).
In my opinion, the paper is well written, contains novel genetic information about Hibiscus sabdariffa L. and deserves publication. However, the authors should make some minor corrections and modifications of the text before it can be accepted for publication. Below I list more detailed comments:
1. Page 1, line 35. Should be: altissima.
2. Page 1, line 37 and many other places in the text. There is no space between last word of the sentence and the citation number or other fragment of the sentence: subtropics[1].
3. Page 1, line 44. Should be introduced from abroad.
4. Page 3, lines 106-110, Table 1. Wrong text formatting. Table’s title is incorrectly placed. Line numbers 111-117 are missing.
5. Page 3, line 127. When Latin name is given at the beginning of a sentence, it should be given in full.
6. Table 3. At 16 of the 32 investigated loci there was heterozygote deficiency compared to the HWE expectations. This situation is commonly observed in studies using SSR markers and is like due to the occurrence of “silent” alleles caused by the lack of amplification of some fragments. The authors should comment this issue.
7. Page 6, line 173. Should be: considerable.
8. Page 6, line 188. Please replace “among” with “from”.
9. Page 8, line 268. Please replace “sliding” with “frameshift”.
10. Page 8, line 27 and many other places in the text. Please give Latin names of other species. Should be: Apocynum (toina).
11. Page 9, line 286. Should be: Hibiscus.
12. Page 12, line 397. Please provide reference for the software Popgene v. 1.32. The same applies to PowerMarker and NTSYS v. 2.1, Nei’s information index. In Table 3 it is called diversity index.
13. Please check carefully the References section. Names of some journal are given in full while others are abbreviated.
Author Response
Response to Reviewer 1 comments
Thank you very much for taking the time to review this manuscript. Please find the detailed responses below and the corresponding revisions highlighted in the re-submitted files.
- Page 1, line 35. Should be: altissima.
Response:Thank you for pointing this out. The wrong character in line 35 has been corrected.
- Page 1, line 37 and many other places in the text. There is no space between last word of the sentence and the citation number or other fragment of the sentence: subtropics [1].
Response: Thanks. Space has been set between last word of the sentence and the citation number in all the manuscript.
- Page 1, line 44. Should be introduced from abroad.
Response: We have revised this issue according to the suggestion.
- Page 3, lines 106-110, Table 1. Wrong text formatting. Table’s title is incorrectly placed. Line numbers 111-117 are missing.
Response: The wrong text format and wrong place have been corrected.
- Page 3, line 127. When Latin name is given at the beginning of a sentence, it should be given in full.
Response: We have given the full Latin name here according to the comment.
- Table 3. At 16 of the 32 investigated loci there was heterozygote deficiency compared to the HWE expectations. This situation is commonly observed in studies using SSR markers and is like due to the occurrence of “silent” alleles caused by the lack of amplification of some fragments. The authors should comment this issue.
Response: We agree with you. Therefore, we have commented the issue according to the comment.
- Page 6, line 173. Should be: considerable.
Response: The word has been corrected.
- Page 6, line 188. Please replace “among” with “from”.
Response: We have replaced “among” with “from”.
- Page 8, line 268. Please replace “sliding” with “frameshift”.
Response: The word “sliding” has been replaced with “frameshift”.
- Page 8, line 27 and many other places in the text. Please give Latin names of other species. Should be: Apocynum (toina).
Response: Thank you for the suggestion. We have supplemented the Latin names of species mentioned in our manuscript.
- Page 9, line 286. Should be: Hibiscus.
Response: The word has been corrected.
- Page 12, line 397. Please provide reference for the software Popgene v. 1.32. The same applies to PowerMarker and NTSYS v. 2.1, Nei’s information index. In Table 3 it is called diversity index.
Response: Thank you. We have provided the website address for each software mentioned in this part.
- Please check carefully the References section. Names of some journal are given in full while others are abbreviated.
Response: Thank you so much. We have checked the References section carefully and the names of all the journals are abbreviated.
Reviewer 2 Report
Comments and Suggestions for Authors
The authors present a study focusing on developing and evaluating simple sequence repeat (SSR) markers derived from the transcriptome of Hibiscus sabdariffa L. However, the study could offer valuable insights into genetic diversity and molecular breeding for roselle following improvement.
- The authors should consider expanding on the significance of SSR markers in genetic diversity studies beyond the Malvaceae family to include a broader context, in the Introduction section.
- In the Results section, further clarify the parameters for selecting highly polymorphic markers to help readers understand the threshold values used.
- A few references are outdated. Including recent studies on SSR marker development using transcriptome sequencing in other crops could make the discussion more relevant.
- The genetic diversity analysis would benefit from more discussion on how the observed diversity might impact breeding programs. Providing recommendations on how these markers could be applied to specific breeding goals would enhance the study's applicability.
- Some sections, such as the primer selection process and polymorphism validation, would benefit from a more detailed explanation to underscore their significance.
- Overall, the MS would benefit from further editing to correct grammatical errors and improve readability.
Needs to be improved.
Author Response
We really appreciate you for taking the time to review this manuscript. Please find the detailed responses below and the corresponding corrections highlighted in the re-submitted files.
Comments and Suggestions for Authors
The authors present a study focusing on developing and evaluating simple sequence repeat (SSR) markers derived from the transcriptome of Hibiscus sabdariffa L. However, the study could offer valuable insights into genetic diversity and molecular breeding for roselle following improvement.
The authors should consider expanding on the significance of SSR markers in genetic diversity studies beyond the Malvaceae family to include a broader context, in the Introduction section.
Response: Thank you so much for your suggestion. We have expanded on the significance of SSR markers in genetic diversity studies beyond the Malvaceae family in the Introduction part (line 55-61).
In the Results section, further clarify the parameters for selecting highly polymorphic markers to help readers understand the threshold values used.
Response: We agree with you. We have clarified the parameters for selecting highly polymorphic markers in Results section (line 147-150), so the readers can understand the threshold values used more better.
A few references are outdated. Including recent studies on SSR marker development using transcriptome sequencing in other crops could make the discussion more relevant.
Response: We have supplemented recent studies on SSR marker development using transcriptome sequencing in other crops, which should make the discussion more relevant (line 247-253).
The genetic diversity analysis would benefit from more discussion on how the observed diversity might impact breeding programs. Providing recommendations on how these markers could be applied to specific breeding goals would enhance the study's applicability.
Response: Thank you for pointing this out. We have revised this issue according to the suggestion (line 311-321).
Some sections, such as the primer selection process and polymorphism validation, would benefit from a more detailed explanation to underscore their significance.
Response: Thank you for your suggestion. This issue has been revised in “SSR primer validation” part according to the comment.
Overall, the MS would benefit from further editing to correct grammatical errors and improve readability.
Response: The MS has been edited by the professional English language editor, and we believe the improvement would make its readability much better.
Reviewer 3 Report
Comments and Suggestions for Authors
Title: Development of roselle (Hibiscus sabdariffa L.) transcriptome-based simple sequence repeat markers and their application in roselle
The study describes the development of SSR markers for Hibiscus sabdariffa, using RNA sequencing technology, where 32 polymorphic loci were selected to quantify genetic diversity, population structure and DNA fingerprint in the species. This battery of loci was validated using a sample of 38 individuals, where 33 samples were H. sabdariffa var. sabdariffa and 5 were H. sabdariffa var. altissima. The authors estimated the genetic diversity by loci and average between loci for the sample.
However, there is a lack of a linkage equilibrium test between loci to verify whether the association between alleles of different loci is random or not. This information is important when we develop new genetic markers, since pairs of loci showing strong linkage disequilibrium (non-random association between alleles of different loci) violate the assumption of independent segregation between alleles of different loci. This is especially important in breeding system analyses, paternity analyses, kinship estimates, among others. In such cases, one of the loci of the pair showing disequilibrium should be eliminated from the analysis to avoid bias in the estimates.
Therefore, I suggest that the authors add this analysis to the study.
Additionally, I have some suggestions and corrections to improve the current version of the manuscript.
Minor corrections
Lines 32-33. Where you read …..Hibiscus sabdariffa var. sabdariffa (HSS) and Hibiscus sabdariffa var. altissima (HSA)….. the correc is … …..Hibiscus sabdariffa var. sabdariffa (HSS) and Hibiscus sabdariffa var. altissima (HSA)…..
Please, check this and correct the entire manuscript.
Lines 52-53. Where you read ….. investigations are inefficient, expensive, and are inefficient for the assessment….. I suggest … investigations are expensive and inefficient for the assessment…..
Lines 61-63. Where you read ….. Furthermore, SSR markers are recommended by the International Union for the Protection of New Plant Varieties as molecular markers for constructing DNA fingerprint databases [18]….
This statement is not entirely true. In fact, we are using SNP markers for DNA fingerprinting because they are more stable, high frequency in the genomes, and allow us to obtain hundreds of loci.,1,184
So, I suggest … Furthermore, SSR markers together to SNPs markers are…
Line 83. Where you read ….. tool for diversity analysis, variety …..I suggest … ….. tool for genetic diversity analysis, variety…..
Author Response
We are so grateful to you for taking the time to review this manuscript. Please find the detailed responses below and the corresponding revisions highlighted in the re-submitted files.
The study describes the development of SSR markers for Hibiscus sabdariffa, using RNA sequencing technology, where 32 polymorphic loci were selected to quantify genetic diversity, population structure and DNA fingerprint in the species. This battery of loci was validated using a sample of 38 individuals, where 33 samples were H. sabdariffa var. sabdariffa and 5 were H. sabdariffa var. altissima. The authors estimated the genetic diversity by loci and average between loci for the sample.
However, there is a lack of a linkage equilibrium test between loci to verify whether the association between alleles of different loci is random or not. This information is important when we develop new genetic markers, since pairs of loci showing strong linkage disequilibrium (non-random association between alleles of different loci) violate the assumption of independent segregation between alleles of different loci. This is especially important in breeding system analyses, paternity analyses, kinship estimates, among others. In such cases, one of the loci of the pair showing disequilibrium should be eliminated from the analysis to avoid bias in the estimates.
Therefore, I suggest that the authors add this analysis to the study.
Response: Thank you for your comment, and this is a meaningful suggestion. However, studies have shown that only those SSR markers with known chromosomal information can be used for linkage equilibrium test [1]. Unfortunately, there is no genome at the chromosomal level for roselle till now, and therefore, it is not possible to estimate the linkage equilibrium value of SSR loci in this research. Hopefully, we can get the information of a chromosomal genome for roselle in the future. Our results show the screened SSR markers were able to distinguish the 38 roselle accessions clearly, and which fingerprints were constructed with 5 pairs of SSR primers, so it indicated the SSR markers in this study are reliable.
[1] P Bai, X Z Cheng, L X Wang, S H Wang and H L Chen. Genetic diversity, population structure and linkage disequilibrium in Adzuki bean by using SSR markers. ACTA AGRONOMICA SINICA 2014, 40(5):788-797.
Minor corrections
Lines 32-33. Where you read …..Hibiscus sabdariffa var. sabdariffa (HSS) and Hibiscus sabdariffa var. altissima (HSA)….. the correc is … …..Hibiscus sabdariffa var. sabdariffa (HSS) and Hibiscus sabdariffa var. altissima (HSA)…..
Please, check this and correct the entire manuscript.
Response: Thanks. We checked and corrected this through entire manuscript.
Lines 52-53. Where you read ….. investigations are inefficient, expensive, and are inefficient for the assessment….. I suggest … investigations are expensive and inefficient for the assessment…..
Response: We deleted the first “inefficient” in the sentence.
Lines 61-63. Where you read ….. Furthermore, SSR markers are recommended by the International Union for the Protection of New Plant Varieties as molecular markers for constructing DNA fingerprint databases [18]….
This statement is not entirely true. In fact, we are using SNP markers for DNA fingerprinting because they are more stable, high frequency in the genomes, and allow us to obtain hundreds of loci.,1,184
So, I suggest … Furthermore, SSR markers together to SNPs markers are…
Response: We agree with your suggestion, and we revised this part according to the comment.
Line 83. Where you read ….. tool for diversity analysis, variety …..I suggest … ….. tool for genetic diversity analysis, variety…..
Response: Thank you. The word “diversity” was corrected to “genetic diversity” based on the suggestion.